# Technology to Automatically Record Eating Behavior in Real Life: A Systematic Review

**DOI:** 10.3390/s23187757

**Published:** 2023-09-08

**Authors:** Haruka Hiraguchi, Paola Perone, Alexander Toet, Guido Camps, Anne-Marie Brouwer

**Affiliations:** 1TNO Human Factors, Netherlands Organization for Applied Scientific Research, Kampweg 55, 3769 DE Soesterberg, The Netherlandsanne-marie.brouwer@tno.nl (A.-M.B.); 2Kikkoman Europe R&D Laboratory B.V., Nieuwe Kanaal 7G, 6709 PA Wageningen, The Netherlands; 3Division of Human Nutrition and Health, Wageningen University & Research, Stippeneng 4, 6708 WE Wageningen, The Netherlands; 4OnePlanet Research Center, Plus Ultra II, Bronland 10, 6708 WE Wageningen, The Netherlands; 5Department of Artificial Intelligence, Radboud University, Thomas van Aquinostraat 4, 6525 GD Nijmegen, The Netherlands

**Keywords:** eating, drinking, daily life, real life, sensors, technology, behavior

## Abstract

To monitor adherence to diets and to design and evaluate nutritional interventions, it is essential to obtain objective knowledge about eating behavior. In most research, measures of eating behavior are based on self-reporting, such as 24-h recalls, food records (food diaries) and food frequency questionnaires. Self-reporting is prone to inaccuracies due to inaccurate and subjective recall and other biases. Recording behavior using nonobtrusive technology in daily life would overcome this. Here, we provide an up-to-date systematic overview encompassing all (close-to) publicly or commercially available technologies to automatically record eating behavior in real-life settings. A total of 1328 studies were screened and, after applying defined inclusion and exclusion criteria, 122 studies were included for in-depth evaluation. Technologies in these studies were categorized by what type of eating behavior they measure and which type of sensor technology they use. In general, we found that relatively simple sensors are often used. Depending on the purpose, these are mainly motion sensors, microphones, weight sensors and photo cameras. While several of these technologies are commercially available, there is still a lack of publicly available algorithms that are needed to process and interpret the resulting data. We argue that future work should focus on developing robust algorithms and validating these technologies in real-life settings. Combining technologies (e.g., prompting individuals for self-reports at sensed, opportune moments) is a promising route toward ecologically valid studies of eating behavior.

## 1. Introduction

As stated by the World Health Organization (WHO) “Nutrition is coming to the fore as a major modifiable determinant of chronic disease, with scientific evidence increasingly supporting the view that alterations in diet have strong effects, both positive and negative, on health throughout life” [1]. It is therefore of key importance to find efficient and solid methodologies to study eating behavior and food intake in order to help reduce potential long-term health problems caused by unhealthy diets. Past research on eating behaviors and attitudes relies intensively on self-reporting tools, such as 24-h recalls, food records (food diaries) and food frequency questionnaires (FFQ; [2,3,4]). However, there is an increasing understanding of the limitations of this classical approach to studying eating behaviors and attitudes. One of the major limitations of this approach is that self-reporting tools rely on participants’ recall, which may be inaccurate or biased (especially when studying the actual amount of food or liquid intake [5]). Recall biases can be caused by demand characteristics, which are cues that may indicate the study aims to participants, leading them to change their behaviors or responses based on what they think the research is about [6], or more generally by the desire to comply with social norms and expectations when it comes to food intake [7,8]. Additionally, the majority of the studies investigating eating behavior are performed in the lab, which does not allow for a realistic replication of the many influences on eating behavior that occur in real life (e.g., [9]). To overcome these limitations, it is crucial to examine eating behavior and the effect of interventions in daily life, at home or at institutions such as schools and hospitals. In contrast to lab research settings, humans typically behave naturally in these settings. It is also important that testing real-life eating behaviors in naturalistic settings relies on implicit, nonobtrusive measures [10] that are objective and able to overcome potential biases.

There is growing interest in identifying technologies able to improve the quality and validity of data collected to advance nutrition science. Such technologies should enable eating behavior to be measured passively (i.e., without requiring action or mental effort on the part of the users), objectively and reliably in realistic contexts. To maximize the efficiency of real-life measurement, it is vital to develop technologies that capture eating behavior patterns in a low-cost, unobtrusive and easy-to-analyze way. For real-world practicality, the technologies should be comfortable and acceptable so that they can be used in naturalistic settings for extended periods while respecting the users’ privacy.

To gain insight into the state of the art in this field, we performed a search for published papers using technologies to measure eating behavior in real-life settings. In addition to papers describing specific systems and technologies, this search returned many review papers, some of which contained systematic reviews.

Evaluating these systematic reviews, we found that an up-to-date overview encompassing all (close-to) available technologies to automatically record eating behavior in real-life settings is still missing. To fill this gap, we here provide such an overview, categorized by what type of eating behavior they measure and which type of sensor technology they use. We indicate to what extent these technologies are readily available for use. With this review, we aim to (1) help researchers identify the most suitable technology to measure eating behavior in real life and to provide a basis for determining the next steps in (2) research on measuring eating behavior in real life and (3) technology development.

## 2. Methods and Procedure

### 2.1. Literature Review

Our literature search reporting adheres to the Preferred Reporting Items for Systematic reviews and Meta-Analyses (PRISMA) checklist [11,12]. The PRISMA guidelines ensure that the literature is reviewed in a standard and systematic manner. This process underlies four phases: identification, screening, eligibility and inclusion. The PRISMA diagram showing the search flow and inclusion/exclusion of records and reports in this study is shown in Figure 1.

### 2.2. Eligibility Criteria

Our literature search aimed to identify mature and practical (i.e., not too complex or restrictive) technologies that can be used to unobtrusively assess food or drink intake in real-life conditions. The inclusion criterium was a sensor-based approach to the detection of eating or drinking. Studies not describing a sensor-based device to detect eating or drinking were excluded.

### 2.3. Information Sources and Search

The literature search included two stages: an initial automatic search of online databases and a manual search based on the reference lists from the papers selected from the previous search stage (using a snowballing approach [13,14]). In addition, papers recommended by a reviewer were added.

The initial search was conducted on 24 February 2023 across the ACM Digital Library, Google Scholar (first 100 results), IEEE Xplore, MDPI, PubMed and Scopus (Elsevier) databases. The results were date-limited from 2018 to date (i.e., 24 February 2023). As this field is rapidly advancing and sensor-based technologies evaluated in earlier papers are likely to have been further developed, we initially limited our search to the past 5 years.

Our broad search strategy was to identify papers that included terms in their title, abstract or keywords related to food and drink, eating or drinking activities and the assessment of the amount consumed.

The search was performed using equivalents of the following Boolean search string (where* represents a wildcard): “(beverage OR drink OR food OR meal) AND (consum* OR chew* OR eating OR ingest* OR intake OR swallow*) AND (portion OR serving OR size OR volume OR mass OR weigh*) AND (assess OR detect OR monitor OR measur*)”. Some search terms were truncated in an effort to include all variations of the word.

The records retrieved in the initial search across the six databases and the results of manual bibliography searches were imported into EndNote 20 (www.endnote.com) and duplicates were removed.

### 2.4. Screening Strategy

Figure 1 presents an overview of the screening strategy. In the first round, the titles and abstracts returned (*n* = 1241, after the elimination of 68 duplicates) were reviewed against the eligibility criterium. If the title and/or abstract mentioned a sensor-based approach to the detection of eating or drinking, the paper was included in the initial screening stage to be further assessed in the full-text screening stage. Papers that did not describe a sensor-based device to detect eating or drinking were excluded. Full-text screening was conducted on the remaining articles (*n* = 126), leading to a final sample of 73 included papers from the initial automatic search. Papers focusing on animal studies (*n* = 4), food recognition (*n* = 7), nutrient estimation (*n* = 6), system design (*n* = 2) or other nonrelated topics (*n* = 10) were excluded. While review papers (*n* = 20) were also excluded from our technology overview (Table 1), they were evaluated (Table A1 and Table A2 in Appendix A) and used to define the scope of this study. Additional papers were identified by manual search via the reference lists of full texts that were screened (*n* = 91). Full-text screening of these additional papers led to a final sample of 49 included papers from the manual search. Papers about dietary recall (*n* = 7), food recognition (*n* = 11), nutrient estimation (*n* = 7), describing systems already described in papers from the initial automatic search (*n* = 2) or other nonrelated topics (*n* = 6) were excluded. Again, review papers (*n* = 8) were excluded from our technology overview (Table 1) but evaluated and used to define the scope of this study. The total number of papers included in this review amounts to 123.

All screening rounds were conducted by two of the authors. Each record was reviewed by two reviewers to decide its eligibility based on the title and abstract of each study, taking into consideration the exclusion criteria. When a record was rejected by one reviewer and accepted by the other, it was further evaluated by all authors and kept for eligibility when a majority voted in favor.

### 2.5. Reporting

We evaluated and summarized the review papers that our search returned in two tables (Appendix A). Table A1 includes systematic reviews, while Table A2 includes nonsystematic reviews. We defined systematic reviews as reviews following the PRISMA methodology. For all reviews, we reported the year of publication and the general scope of the review. For systematic reviews, we also reported the years of inclusion, the number of papers included and the specific requirements for inclusion.

We summarized the core information about the devices and technologies for measuring eating and drinking behaviors from our search results in Table 1. This table categorizes the studies retrieved in our literature search in terms of their measurement objectives, target measures, the devices and algorithms that were used as well as their (commercial or public) availability and the way they were applied (method). In the column “Objective”, the purposes of the measurements are described. The three objectives we distinguish are “Eating/drinking activity detection”, “Bite/chewing/swallowing detection” and “Portion size estimation”. Note that the second and third objectives can be considered as subcategories of the first—technologies are included in the first if they could not be grouped under the second or third objective. The objectives are further specified in the column “Measurement targets”. In the column “Device”, we itemize the measurement tools or sensors used in the different systems. For each type of device, one or more representative papers were selected, bearing in mind the TRL (technology readiness level [15]), the availability (off-the-shelf) of the device and algorithm that were used, the year of publication (recent) and the number of times it was cited. The minimum TRL level was 2 and the paper with the highest TRL level among papers using similar techniques was selected as the representative paper. A concise description of each representative example is given in the “Method” column. The commercial availability of the example devices and algorithms is indicated in the “Off-the-shelf device” and “Ready-to-use algorithm” columns. Lastly, other studies using similar systems are listed in the “Similar papers” column. Systems combining devices for several different measurement targets can appear in different table rows. To indicate this, they are labeled with successive letters for each row they appear in (e.g., 1a and 1b).

For each of the three objectives, we counted the number of papers that described sensors that are designed (1) to be attached to the body, (2) to be attached to an object, (3) to be placed in the environment or (4) to be held in the hand. Sensors attached to the body were further subdivided by body location. The results are visualized using bar graphs.

## 3. Results

Table 1 summarizes the core information of devices and technologies for measuring eating and drinking behaviors from our search results.

### 3.1. Eating and Drinking Activity Detection

For “eating/drinking activity detection”, many systems have been reported that measure eating- and drinking-related motions. In particular, many papers reported measuring these actions using motion sensors such as inertial sensor modules (i.e., inertial measurement units or IMUs). IMUs typically consist of various sensors such as an accelerator, gyroscope and magnetometer. These sensors are embedded in smartphones and wearable devices such as smartwatches. In [16], researchers collected IMU signals with off-the-shelf smartwatches to identify hand-based eating and drinking-related activities. In this case, participants wore smartwatches on their preferred wrists. Other studies have employed IMUs worn on the wrist, upper arm, head, neck and combinations thereof [17,18,19,20]. IMUs worn on the wrist or upper arms can collect movement data relatively unobtrusively during natural eating activities such as lifting food or bringing utensils to the mouth. Recent research has also improved IMUs that are attached to the head or neck, combining sensors with glasses or necklaces so that they are less bulky and users are not aware that they are being worn. Besides IMUs, proximity sensors, piezoelectric sensors and radar sensors are also used to detect hand-to-mouth gestures or jawbone movements [21,22,23]. Pressure sensors are used to measure eating activity as well. For instance, in [24], eating activities and the amount of consumed food are measured by a pressure-sensitive tablecloth and tray. These devices provide information on food-intake-related actions such as cutting, scooping, stirring or the identification of the plate or container on which the action is executed and allow the tracking of weight changes of plates and containers. Microphones, RGB-D images and video cameras are also used to detect eating and drinking-related motions. In [25], eating actions are detected by a ready-to-use algorithm as the 3D overlap of the mouth and food, using RGB-D images taken with a commercially available smartphone. These imaging techniques have the advantage of being less physically intrusive than wearable motion sensors but they still restrict subjects’ natural eating behavior as the face must be clearly visible to the camera. Ear-worn sensors can measure in-body glucose levels [26] and tooth-mounted dielectric sensors can measure impedance changes in the mouth signaling the presence of food [27]. Although these latter methods can directly detect eating activity, the associated devices and data processing algorithms are still in the research phase. In addition, ref. [28] reports a wearable array of microneedles for the wireless and continuous real-time sensing of two metabolites (lactate and glucose, or alcohol and glucose) in the interstitial fluid (Figure 2). This is useful for truly continuous, quantitative, real-time monitoring of food and alcohol intake. Future development of the system is needed on several aspects such as battery life and advanced calibration algorithms.

### 3.2. Bite, Chewing or Swallowing Detection

In the “bite/chewing/swallowing detection” category, we grouped studies in which the number of bites (bite count), bite weight and chewing or swallowing actions are measured. Motion sensors and video are used to detect bites (count). For instance, OpenPose is an off-the-shelf software that analyzes bite counts from videos [29]. To assess bite weight, weight sensors and acoustic sensors have been used [30,31]. In [30], the bite weight measurement also provides the estimation of a full portion.

Chewing or swallowing is the most well-studied eating- and drinking-related activity, as reflected by the number of papers focusing on such activities (31 papers). Motion sensors and microphones are frequently employed for this purpose. For instance, in [32], a gyroscope is used for chewing detection, an accelerometer for swallowing detection and a proximity sensor to detect hand-to-mouth gestures. Microphones are typically used to register chewing and swallowing sounds. In most cases, commercially available microphones are applied, while the applied detection algorithms are custom-made. Video, electroglottograph (EGG) and electromyography (EMG) devices are also used to detect chewing and swallowing. EGG detects the variations in the electrical impedance caused by the passage of food during swallowing, while EMG in these studies monitors the masseter and temporalis muscle activation for recording chewing strokes. The advantages of EGG and EMG are that they can directly detect swallowing and chewing while eating and are not, or are less, affected by other body movements compared to motion sensors. However, EMG devices are not wireless and EGG sensors need to be worn around the face, which is not optimal for use in everyday eating situations.

### 3.3. Portion Size Estimation

Portion size is estimated mainly by using weight sensors and food image analysis. Regarding weight sensors, the amount of food consumed is calculated by comparing the weights of plates before and after eating. An open-source system consisting of a wireless pocket-sized kitchen scale connected to a mobile application has been reported in [33]. As shown in Figure 3, a system turning an everyday smartphone into a weighing scale is also available [34]. The relative vibration intensity of the smartphone’s vibration motor and its built-in accelerometer are used to estimate the weight of food that is placed on the smartphone. Off-the-shelf smartphone cameras are typically used for volume estimation from food images. Also, several studies use RGB-D images to get more accurate volume estimations from information on the height of the target food. For image-based approaches, AI-based algorithms are often employed to calculate portion size. Some studies made prototype systems applicable to real-life situations. In [35], acoustic data from a microphone was collected along with food images to measure the distance from the camera to the food. This enables the food in the image to be scaled to its actual size without training images and reference objects. However, in other cases, image processing mostly uses a reference for comparing the food size. Besides image analysis, in [36], researchers took a 360-degree scanned video obtained with a laser module and a diffraction lens and applied their volume estimation algorithm to the data. In addition to the above devices, a method to estimate portion size using EMG has been reported [37]. In this study, EMG embedded in an armband device detects different patterns of signals based on the weight that a user is holding.

For estimating portion size in drinks, several kinds of sensors have been tested. An IMU in a smartwatch was used to estimate drink intake volume from sip duration [38]. Also, in [39], liquid sensors such as a capacitive sensor and a conductivity sensor were used to monitor the filling levels in a cup. Some research groups developed so-called smart fridges that automatically register food items and quantities. In [40], image analysis of a thermal image taken by an infrared (IR) sensor embedded in a fridge provides an estimation of a drink volume. Another study proposed a system called the Playful Bottle system [41], which consists of a smartphone attached to a common drinking mug. Drinking motions such as picking up the mug, tilting it back and placing it on the desk are detected by the phone’s accelerometer. After the drinking action is completed and the water line becomes steady, the phone’s camera captures an image of the amount of liquid in the mug (Figure 4).

### 3.4. Sensor Location

Figure 5 indicates where sensors are typically located per objective. The locations of the sensors are classified as body-attached (e.g., ear, neck, head, glasses), embedded in objects (e.g., plates, cutlery) and the environment (e.g., distant camera, magnetic trackers). For eating/drinking activity detection, sensors are mostly worn on the body, followed by embedded in the objects. Body-worn sensors are also used for bite/chewing/swallowing detection. On the other hand, for portion size estimation, object-embedded and handheld sensors are mainly chosen depending on the measuring targets. Figure 6 shows the locations of wearable body sensors used in the reviewed studies. Sensors attached to wrists are most frequently used (32 cases), followed by embedded in glasses (19 cases) and attached to the ear (14 cases).

**Table 1 sensors-23-07757-t001:** Summary of core information of devices and technologies for measuring eating and drinking behaviors. The commercial availability of the example devices and algorithms is indicated “Y”(Yes) and “N”(No). Letters a and b following the reference numbers indicate systems that combine devices for several different measurement targets, and therefore appear in two table rows (i.e., a and b).

Objective	Measurement Target	Device	Representative Paper	Method	Off-the-Shelf Device	Ready-to-Use Algorithm	Similar Papers
Eating/drinking activity detection	eating/drinking motion	motion sensor	[16]	eating and drinking detection from smartwatch IMU signal	Y	N	[42] a, [43] a, [44,45,46], [47] a, [41] a, [21,48], [49] a, [22,50,51], [52] a, [38] a, [53] a, [17,18,54,55], [37] a, [56], [19] a, [57,58,59], [60] a, [61], [62] a, [63], [20] a, [64] a, [42,65,66,67,68]
			[23]	detecting eating and drinking gestures from FMCW radar signal	N	N	
			[24] a	eating activities and amount consumed measured by pressure-sensitive tablecloth and tray	N	N	
		microphone	[47] b	eating detection from fused inertial-acoustic sensing using smartwatch with embedded IMU and microphone	Y	N	[26] a, [69], [60] b
		RGB-D image	[25] a	eating action detected from smartphone RGB-D image as 3D overlap between mouth and food	Y	Y	
		video	[70]	eating detection from cap-mounted video camera	Y	N	[55] a
	liquid level	liquid sensor	[71]	capacitive liquid level sensor	N	N	[72]
	impedance change in mouth	dielectric sensor	[27]	RF coupled tooth-mounted dielectric sensor measures impedance changes due to food in mouth	N	N	
	in-body glucose level	glucose sensor	[26] b	glucose level measured by ear-worn sensor	N	N	[28] a
	in-body alcohol level	microneedle sensor	[28] b	alcohol level measured by microneedle sensor on the upper arm	N	N	
	user identification	PPG (photoplethysmography) sensor	[53] b	sensors on water bottle to identify the user from heart rate	N	N	
Bite/chewing/swallowing detection	bites (count)	motion sensor	[73]	a gyroscope mounted on a finger to detect motions of picking up food and delivering it to the mouth	Y	N	[74,75]
		video	[29]	bite count by video analysis using OpenPose pose estimation software	Y	Y	
	bite weight	weight sensor	[30] a	plate-type base station with embedded weight sensors to measure amount and location of bites	N	N	[55] a
		acoustic sensor	[31]	commercial earbuds, estimation model based on nonaudio and audio features	Y	N	
	chewing/swallowing	motion sensor	[32] a	chewing detection from gyroscope, swallowing detection from accelerometer, hand-to-mouth gestures from proximity sensor	Y	Y	[76,77,78,79], [49] b, [80,81], [82] a, [83], [84] a, [85,86,87], [62] b, [20] b
		microphone	[88]	wearable microphone with minicomputer to detect chewing/swallowing sounds	Y	N	[43] b, [89,90,91], [82] b, [19] b, [92], [84] b, [93]
		video	[94]	classification of facial action units related to chewing from video	Y	N	[55] b
		EGG	[95]	swallowing detected by larynx-mounted EGG device	Y	N	
		EMG	[96]	eyeglasses equipped with EMG to monitor temporalis muscles’ activity	N	N	[43] c, [97]
Portion size estimation	portion size food	motion sensor	[34]	acceleration sensor of smartphone, measuring vibration intensity	Y	Y	[98] a
		weight sensor	[33]	wireless pocket-sized kitchen scale connected to app	Y	Y	[99,100,101,102,103], [55] b, [104], [30] b, [98] b, [105], [106] a, [107], [64] b, [24] b
		image	[108]	AI-based system to calculate food leftovers	Y	Y	[32] b, [35,109,110,111,112,113,114,115,116], [117] b, [106] b, [106,118,119,120,121,122,123]
			[35]	measuring the distance from the camera to the food using smartphone images combined with microphone data	Y	N	[124]
			[125]	RGB-D image and AI-based system to estimate consumed food volume using before- and after-meal images	Y	Y	[25] b, [126,127,128,129,130,131,132]
		laser	[36]	360-degree scanned video; the system design includes a volume estimation algorithm and a hardware add-on that consists of a laser module and a diffraction lens	N	N	[133]
		EMG	[37] b	weight of food consumed from EMG data	N	N	
	portion size drink	motion sensor	[38] b	volume from sip duration from IMU in smartwatch	Y	N	[42] b, [52] b
		infrared (IR) sensor	[40]	thermal image by IR sensor embedded in smart fridge	N	N	
		liquid sensor	[39]	capacitive sensor, conductivity sensor, flow sensor, pressure sensor, force sensors embedded in different mug prototypes	N	N	
		image	[41] b	smartphone camera attached to mug	N	N	[134]

## 4. Discussion

This systematic review provides an up-to-date overview of all (close-to) available technologies to automatically record eating behavior in real life. Technologies included in this review should enable eating behavior to be measured passively (i.e., without users’ active input), objectively and reliably in realistic contexts to avoid reliance on subjective user recall. We performed our review in order to help researchers identify the most suitable technology to measure eating behavior in real-life settings and to provide a basis for determining the next steps in both technology development and the measurement of eating behavior in real life. In total, 1332 studies were screened and 123 studies were included after the application of objective inclusion and exclusion criteria. A total of 26 studies contained more than one technology. We found that often, relatively simple sensors are used to measure eating behaviors. Motion sensors are commonly used for eating/drinking activity detection and bite/chewing/swallowing; in addition, microphones are often used in studies focusing on chewing/swallowing. These sensors are usually attached to the body, in particular to the wrist for eating/drinking activity detection and to areas close to the face for detecting bite/chewing/swallowing. For portion size estimation, weight sensors and images from photo cameras are mostly used.

Concerning the next steps in technology development, the information from the “Off-the-shelf device” and “Ready-to-use algorithm” columns in the technology overview table indicates which devices and algorithms are not ready for use yet and would benefit from further development. The category “portion size estimation” seems the most mature with respect to off-the-shelf availability and ready-to-use algorithms. Overall, what is mostly missing is ready-to-use algorithms. It is an enormous challenge to build fixed algorithms that accurately recognize eating behavior under varying conditions of sensor noise, types of food and individuals’ behavior and appearance. Typically, with algorithms, we refer to machine learning or AI algorithms. These are trained using annotated (correctly labeled) data and only work well in conditions that are similar to the ones they were trained in. In most reviewed studies, demonstrations of algorithms are limited to controlled conditions and a small number of participants. Therefore, these algorithms still need to be tested and evaluated for accuracy and generalizability outside the laboratory, such as in homes, restaurants and hospitals.

When it comes to real-life studies, the obtrusiveness of the devices is an important factor. Devices should minimally interfere with the natural behavior of participants. Devices worn on the body with wires connected to a battery or other devices may restrict eating motions and constantly remind participants that they are being recorded. Wireless devices are suitable in that perspective but at the same time, battery duration may be a limitation for long-term studies. Devices such as tray-embedded sensors and cameras that are not attached to the participant’s body are advantageous in terms of both obtrusiveness and battery duration.

Although video cameras can provide holistic data on participants’ eating behaviors, they present privacy concerns. When a camera is used to film the course of a meal, the data provide the participant’s physical characteristics and enable the identification of the participant. Also, when the experiments are performed at home, participants cannot avoid showing their private environment. Ideally, the experiments should allow data to be collected anonymously if this information is not needed for a certain purpose such as clinical data collection. This could be achieved by only storing extracted features from the camera data rather than the images themselves, though this prohibits later validation and improvement of feature extraction [135]. Systems using weight sensors do not suffer from privacy issues as camera images from the face do. Ref. [106] used a weight sensor in combination with a camera pointing downward at the scales to keep track of the consumption of various types of seasonings.

For future research, we think it will be powerful to combine methods and sensor technologies. While most studies rely on single types of technologies, there are successful examples of combinations that illustrate a number of ways in which system and data quality can be improved. For instance, a novel and robust device called SnackBox [136] consists of three containers for different types of snacks embedded on weight sensors (Figure 7) and can be used to monitor snacking behavior at home. It can be connected to wearables and smartphones, thereby allowing for contextualized interpretation of signals recorded from the participant and for targeted ecological momentary assessment (EMA [137]). With EMA, individuals are probed to report current behavior and experiences in their natural environment and this avoids reliance on memory. For instance, when the SnackBox detects snacking behavior, EMA can assess the individual’s current mood state through a short questionnaire. This affords the collection of more detailed and more accurate information compared to asking for this information at a later moment in time. Combining different sensor technologies can also have other benefits. Some studies used a motion detector or an audio sensor as switches to turn on other devices such as a chewing counter or a camera [62,91]. These systems are useful to collect data only during meal durations, thereby limiting superfluous data collection that is undesirable from the point of view of privacy and battery life of the devices that are worn the whole day. In a study imitating a restaurant setting, a system consisting of custom-made table-embedded weight sensors and passive RFID (radio-frequency identification) antennas was used [99]. This system detects the weight change in the food served on the table and recognizes what the food is using RFID tags, thereby complementing information that would have been obtained by using either or alone and facilitating the interpretation of the data. Other studies used an IMU in combination with a microphone to detect eating behaviors [60,82]. It was concluded that the acoustic sensor in combination with motion-related sensors improved the detection accuracy significantly compared to motion-related sensors alone.

Besides investing in research on combining methods and sensor technologies, research applying and validating these technologies in out-of-the-lab studies is essential. Test generalizability between lab and real-life studies should be examined as well as generalizations across situations, user groups and user experience. These studies will lead to further improvements and/or insight into the context in which the technology can or cannot be used.

The current review has some limitations. First, we did not include a measure of accuracy or reliability of the technologies in our table. Some of the reviews listed in our reviews’ table (Table A1 in Appendix A, e.g., [138,139]) included the presence of evaluation metrics indicating the performance of the technologies (e.g., accuracy, sensitivity and precision) as an inclusion criterion. We decided not to have this specific inclusion criterion as we think in our case it is hard to have comparable measures among studies. Also, whether accuracy is “good” very much depends on the specific research question and study design. Second, our classification of whether an algorithm is ready-to-use could not be based on information directly provided in the paper, but should be considered as a somewhat subjective estimate from the authors of this review.

In conclusion, there are some promising devices for measuring eating behavior in naturalistic settings. However, it will take some time before some of these devices and algorithms will become commercially available due to a lack of examples from a large number of test users and in various conditions. Until then, research in- and outside the lab needs to be carried out using custom-made devices and algorithms and/or with combinations of existing devices. The approach to combine different technologies is recommended as it can lead to multimodal datasets consisting of different aspects of eating behavior (e.g., when people are eating and at what rate), dietary intake (e.g., what people are eating and how much) and contextual factors (e.g., why people are eating and with whom). We expect this to result in a much fuller understanding of individual eating patterns and dynamics, in real-time and in context, which can be used to develop adaptive, personalized interventions. For instance, physiological measures reflecting food elicited attention and arousal have been shown to be positively associated with food neophobia (the hesitance to try novel food) [140] in controlled settings [141,142]. Measuring these in naturalistic circumstances together with eating behavior and possible interventions (e.g., based on distracting attention from food) could increase our understanding of food neophobia and inspire methods to stimulate consuming novel, healthy food.

New technologies measuring individual eating behaviors will be beneficial not only in consumer behavioral studies but also in the field of food and medical industries. New insights into eating patterns and traits discovered using these technologies may contribute to clarifying the use of food products in a wide range of consumers or to allow for guidance in improving patients’ diets.

## Figures and Tables

**Figure 1 sensors-23-07757-f001:**
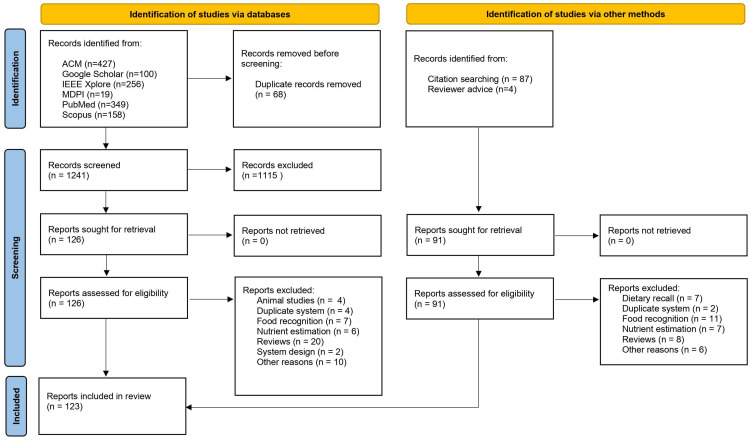
PRISMA flow diagram describing the procedure used to select records and reports for inclusion in this review.

**Figure 2 sensors-23-07757-f002:**
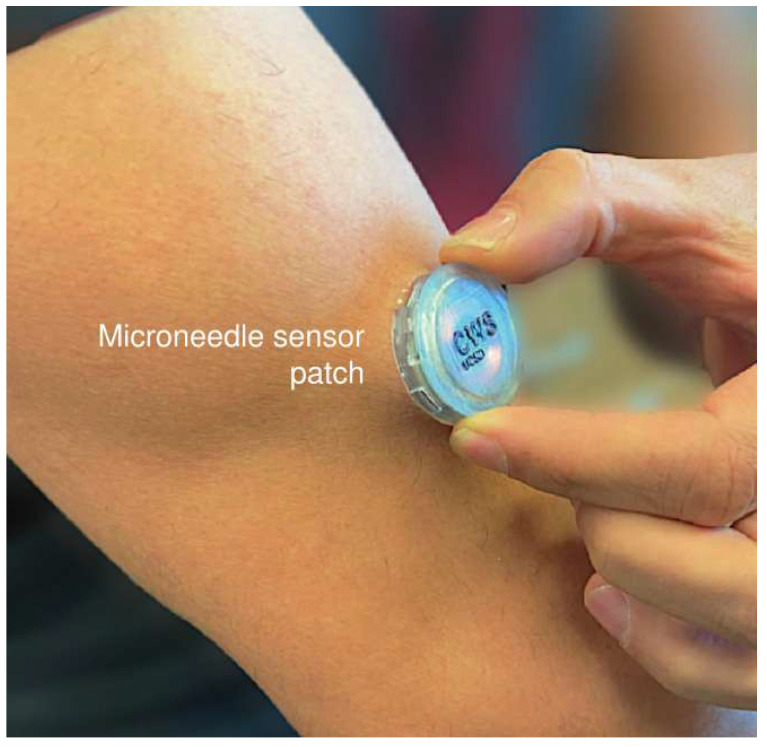
A microneedle-based wearable sensor system (Reprinted with permission from Ref. [28]. 2023, Springer Nature).

**Figure 3 sensors-23-07757-f003:**
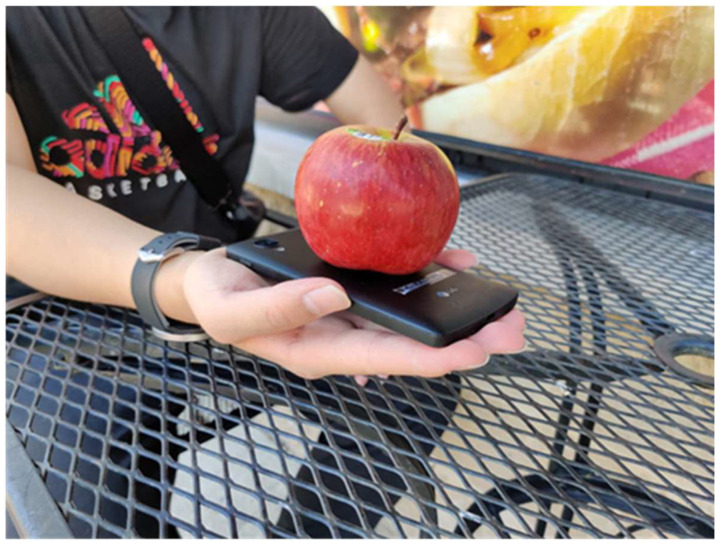
VibroScale (reproduced from [34] with permission).

**Figure 4 sensors-23-07757-f004:**
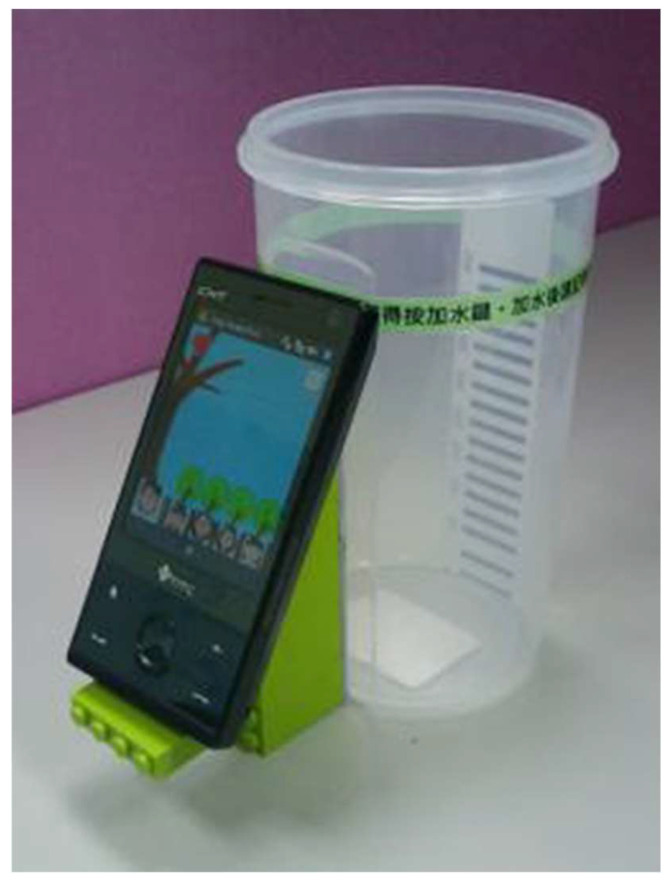
Playful Bottle system (reproduced from [40] with permission).

**Figure 5 sensors-23-07757-f005:**
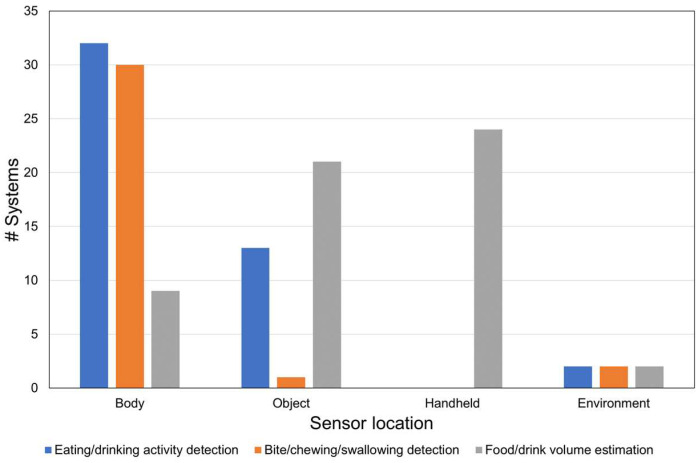
Sensor placement per objective.

**Figure 6 sensors-23-07757-f006:**
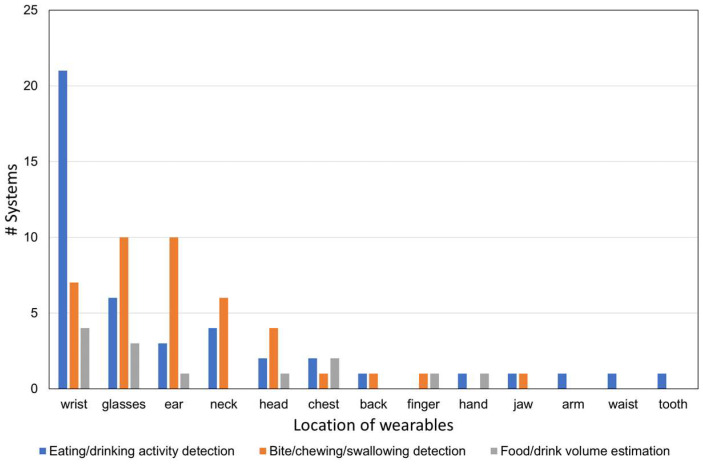
Locations of wearables on the human body.

**Figure 7 sensors-23-07757-f007:**
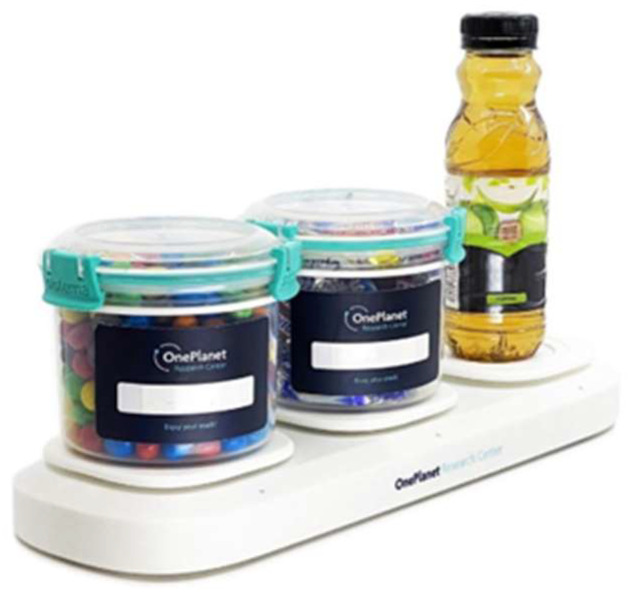
SnackBox (with permission from OnePlanet).

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
