# Peer review of "Technology to Automatically Record Eating Behavior in Real Life: A Systematic Review"

_sensors, 2023, doi:10.3390/s23187757_

Round 1

Reviewer 1 Report

This manuscript will contribute to further ecologically valid studies of eating behavior using sensor technology. This review has structural completeness and systematically organizes and compares the literature in a logical manner, but requires deeper reflection and elaboration of the field and the problems faced. Here are some suggestions for you to consider implementing in the next edition.

1. There are some minor grammatical errors and typos. The paper needs to be revised in terms of language.

2.Figure 1 is not clear and some figures are missing captions.

3.It is recommended that the eating and drinking-activity detection and the biting, chewing, or swallowing detection sections be illustrated with figures, that the studies not be listed, and that a comparative description of the test methods be added.

4.The authors managed to interpret and summarize existing research neutrally, but lacked insightful perspectives.

5.The authors do not appear to have provided a specific description and analysis of the sensor algorithms in each study.

This manuscript will contribute to further ecologically valid studies of eating behavior using sensor technology. This review has structural completeness and systematically organizes and compares the literature in a logical manner, but requires deeper reflection and elaboration of the field and the problems faced. Here are some suggestions for you to consider implementing in the next edition.

1. There are some minor grammatical errors and typos. The paper needs to be revised in terms of language.

2.Figure 1 is not clear and some figures are missing captions.

3.It is recommended that the eating and drinking-activity detection and the biting, chewing, or swallowing detection sections be illustrated with figures, that the studies not be listed, and that a comparative description of the test methods be added.

4.The authors managed to interpret and summarize existing research neutrally, but lacked insightful perspectives.

5.The authors do not appear to have provided a specific description and analysis of the sensor algorithms in each study.

Author Response

We would like to thank the reviewer for reviewing our manuscript and the helpful comments. 

  1. There are some minor grammatical errors and typos. The paper needs to be revised in terms of language.

Response 1: We revised the manuscript using the correction service/software called Grammarly.

  1. Figure 1 is not clear and some figures are missing captions.

Response 2: We made Figure 1 more clear, and added captions for Figure 5 and Figure 7.

  1. It is recommended that the eating and drinking-activity detection and the biting, chewing, or swallowing detection sections be illustrated with figures, that the studies not be listed, and that a comparative description of the test methods be added.

Response 3: A new figure (Figure 2) and related description have been inserted into the ‘eating and drinking-activity detection ’ section [lines 215-220], Table 1, Figure 5 and Figure 6. We considered the technologies included in the ‘biting, chewing, or swallowing ’ section to be relatively easy to imagine from the text, and no figures were added to avoid lengthening the manuscript. We added some comparative descriptions of the test methods in ‘eating and drinking-activity detection ’ section [lines 195-201, 209-211] and ‘biting, chewing, or swallowing ’ section [lines 233-234, 245-249].

  1. The authors managed to interpret and summarize existing research neutrally, but lacked insightful perspectives.

Response 4: In the Results section, we aimed to neutrally summarize and interpret the excisting research; in the Discussion we presented insights such as perspectives on existing challenges and required researchand development avenues. We now also added a part in the Discussion on the value of objective measurement of eating behavior in the context of food neophobia [lines 420-425].

  1. The authors do not appear to have provided a specific description and analysis of the sensor algorithms in each study.

Response 5: This review aims to provide an overall introduction to the state-of-the-art and does not go into each algorithm in depth. This is because most of these algorithms are used in different and very specific environments.

Reviewer 2 Report

Dear Editors:

The article provides a comprehensive review of techniques for studying eating behavior from a variety of sensors such as mechanical and optical sensors that on portable or wearable devices. This review article is in the scope of MDPI Sensors and has good scientific and practical value, so it is recommended for publication. However, the article does not adequately summarize the use of a class of sensors based on body fluid detection for eating behavior, such as alcohol consumption and changes in blood glucose concentration, which are also closely related to eating behavior. The alcohol and glucose sensors can be tested by skin patches or microneedle sensors. Although the authors have cited glucose sensors in the paper, the number is relatively small. It is recommended that the authors add some related research and can refer to the recent research from the group of Joseph Wang, UCSD. and other groups. Here are a few representative articles are given here for reference.

An integrated wearable microneedle array for the continuous monitoring of multiple biomarkers in interstitial fluid. Nat. Biomed. Eng 6, 1214–1224 (2022). https://doi.org/10.1038/s41551-022-00887-1

Wearable and Mobile Sensors for Personalized Nutrition. ACS Sens. 2021, 6 ,5 ,1745–1760. https://doi.org/10.1021/acssensors.1c00553

Leveraging mHealth and Wearable Sensors to Manage Alcohol Use Disorders: A Systematic Literature Review. Healthcare 2022, 10(9), 1672; https://doi.org/10.3390/healthcare10091672

The authors reviewed systematic techniques for studying eating behavior but missed some relative sensor systems such as blood glucose changes, alcohol consumption, and caffeine intake, which are also important to eating behavior study.

.

The further details are in the attached comments file, and I also provide some related references in the comments file that I believe would be helpful.

Author Response

We would like to thank the reviewer for reviewing our manuscript and the helpful comments. A reference to microneedle sensors by Joseph Wang's group is now added to the 'Eating and Drinking-activity Detection' section of the Results [lines 215-220] with a diagram of the sensor (Figure 2), and information about these sensors are added to Table 1, Figure 5 and 6. Other review articles recommended by the reviewer are now included in Appendix Table A1 (systematic reviews) and A2 (non-systematic reviews), in accordance with the procedure described in the ‘Reporting’ section in Methods and Procedure.